# On-Chip Selective Capture and Detection of Magnetic Fingerprints of Malaria

**DOI:** 10.3390/s20174972

**Published:** 2020-09-02

**Authors:** Francesca Milesi, Marco Giacometti, Lorenzo Pietro Coppadoro, Giorgio Ferrari, Gianfranco Beniamino Fiore, Riccardo Bertacco

**Affiliations:** 1Department of Physics, Politecnico di Milano, Piazza Leonardo da Vinci, 32, 20133 Milano, Italy; riccardo.bertacco@polimi.it; 2Department of Electronics Information and Bioengineering, Politecnico di Milano, Via Giuseppe Ponzio, 34, 20133 Milano, Italy; marco.giacometti@polimi.it (M.G.); lorenzopietro.coppadoro@polimi.it (L.P.C.); giorgio.ferrari@polimi.it (G.F.); gianfranco.fiore@polimi.it (G.B.F.); 3IFN-CNR, c/o Politecnico di Milano, Piazza Leonardo da Vinci, 32, 20133 Milano, Italy

**Keywords:** maria diagnosis, diagnostic test, hemozoin, lab-on-chip

## Abstract

The development of innovative diagnostic tests is fundamental in the route towards malaria eradication. Here, we discuss the sorting capabilities of an innovative test for malaria which allows the quantitative and rapid detection of all malaria species. The physical concept of the test exploits the paramagnetic property of infected erythrocytes and hemozoin crystals, the magnetic fingerprints of malaria common to all species, which allows them to undergo a selective magnetophoretic separation driven by a magnetic field gradient in competition with gravity. Upon separation, corpuscles concentrate at the surface of a silicon microchip where interdigitated electrodes are placed in close proximity to magnetic concentrators. The impedance variation proportional to the amount of attracted particles is then measured. The capability of our test to perform the selective detection of infected erythrocytes and hemozoin crystals has been tested by means of capture experiments on treated bovine red blood cells, mimicking the behavior of malaria-infected ones, and suspensions of synthetic hemozoin crystals. Different configuration angles of the chip with respect to gravity force and different thicknesses of the microfluidic chamber containing the blood sample have been investigated experimentally and by multiphysics simulations. In the paper, we describe the optimum conditions leading to maximum sensitivity and specificity of the test.

## 1. Introduction

Malaria, caused by Plasmodium parasites transmitted to people through the bites of Anopheles mosquitoes, is one of the most diffused infectious disease worldwide. The “Word Malaria Report 2019” reports 228 million new cases in 2018, mainly in the African Region (93%), Southeast Asia Region (3.4%), and Eastern Mediterranean Region (2%) [1]. In order to eradicate and eliminate malaria great efforts are needed to improve prevention, diagnosis, and treatment. Apart from traditional methods for prevention, aiming at avoiding the mosquito bites, some vaccines are currently in the trial phase, but their efficacy is still under investigation [2]. On the other hand, malaria treatment is based on well-established drugs but their long-term efficacy strongly depends on a correct usage, limited to true positive cases, to avoid the development of drug resistance. In this sense, there is an emergent need of new diagnostic tests capable to ensure accurate and early stage on-field detection of the infection. Furthermore, highly sensitive point-of-care diagnostic tools could be extremely useful for identifying and treating asymptomatic carriers, in order to block the infection transmission.

Conventional methods for malaria diagnosis are not suitable for on-field wide screening of the population. Optical microscopy examination, the time-honored gold standard for malaria, asks for an expert operator able to distinguish infected red blood cells (i-RBCs) in a blood smear using a good microscope. It is thus an operator-dependent method, requiring approximately 60 min and a laboratory setting [3]. Rapid Diagnostic Tests (RDTs) based on immuno-assay lateral-flow devices require just 15–30 min, but are not quantitative and suffer from a large number of false negative/positive results, especially in endemic zones [4]. Tests based on polymerase chain reaction (PCR) can detect extremely low parasite concentrations, but the on-field compatible version known as Loop-Mediated Isothermal Amplification (LAMP) is not quantitative, has much higher costs, and long operation times (~60 min), which limit the wide spreading of this technology [5].

To fill the gap between the current technology and the need for a cheap, fast, and highly sensitive diagnostic tool suitable to wide screening in low-resource setting, the World Health Organization itself strongly recommends the development of novel RTDs with the same sensitivity of microscopy but with a reduced number of false positives and false negatives with respect to currently available lateral-flow devices.

In this paper, we discuss the capability of a novel rapid diagnostic test under development to selectively detect infected red blood cells (i-RBCs) and free malaria pigment in a blood sample [6]. It is well known from the literature that Plasmodium, during malarial pathogenesis, consumes up to 65% of host red blood cell (RBC) hemoglobin and 80% of its cytoplasm to synthesize amino acids [7,8]. During the digestion of these substances, α-hematin is produced [9]. α-hematin is toxic to both the parasite and the RBC [9], and it is converted into β-hematin crystals, an insoluble biocrystal also called hemozoin crystal (HC) [10,11,12].

β-hematin crystals are made of dimers of hematin linked together by hydrogen bonds. The central Fe${}^{3+}$ ion of each hematin is connected to the oxygen of the carboxylate (RCOO-) side chain of the adjacent hematin through an iron–oxygen coordinate bond. Hemozoin crystals produced by Plasmodium are usually composed by approximately 80,000 heme molecules forming crystals 100–400 nm long with a brick like structure (1:1:8) [13,14]. Noteworthy, the number of the hemozoin crystals in the RBCs depends on the stage of the parasite development: a lower amount of crystals in the ring stage with respect to the schizont stage is found [15].

Interestingly enough, hemozoin has been used as malaria biomarker since the very beginning of microscopy analysis. The detection of free HCs in blood smears by expert microscopists can help to identify the stage of the infection, as large concentrations are expected after the cell membrane rupture, responsible for the febrile attack. Furthermore, free malaria pigment is the unique detectable marker for microscopy in case of sequestration. In particular for *Plasmodium falciparum*, the adherence of infected erythrocytes containing late developmental stages of the parasite (trophozoites and schizonts) to the endothelium of capillaries and venules strongly reduces the concentration of i-RBCs detectable after staining [16].

Furthermore, hemozoin crystals exhibit interesting physical properties such as paramagnetism and optical dichroism. Paramagnetism is given by the central iron ion of hematin in ferric state Fe${}^{3+}$. The ion is in the high-spin configuration due to five unpaired electrons 3d${}^{5}$ resulting in a spin angular momentum S = 5/2, thus owning paramagnetic behavior [6,17,18]. After the conversion of hemoglobin into hemozoin, the latter is expelled by the plasmodium inside the host RBC. Consequently, infected red blood cells (i-RBCs) exhibit paramagnetic properties depending on their crystals content. In the last year, many different approaches to malaria diagnosis based on malaria biomarkers have been proposed [19,20,21,22,23,24,25].

Our test is based on paramagnetic properties of malaria i-RBCs and HCs which allow their selective magnetophoretic separation, driven by an high magnetic field gradient, in a lab-on-chip platform. The infected blood corpuscles are concentrated at the surface of gold interdigitated electrodes thanks to to magnetic concentrators placed in closed proximity whereas healthy ones sediment under the action of gravity. Then, a change in resistivity, proportional to the amount of attracted particles (i-RBCs and HCs) is detected as an impedance variation.

Here, we show that in the horizontal configuration (chip surface parallel to the floor) our system selectively detects only hemozoin crystals, while in the vertical configuration (chip surface perpendicular to the floor), also i-RBCs are sorted and detected. In particular, the impact on the detection efficiency of the angle α between the chip normal and gravity force, as well as of the height of the microfluidic cell above the chip is discussed.

## 2. Materials and Methods

### 2.1. Measurement Setup

Following a lab-on-chip approach, the test is based on a silicon microchip (Figure 1e) that has been fabricated in Polifab, the micro- and nanotechnology center of the Politecnico di Milano. The process flow for the microchip fabrication can be divided in two parts: (i) the realization of an hexagonal lattice of nickel (Ni) microconcentrators, which enables the magnetophoretic separation and so the attraction of both i-RBCs and free HCs on top of them, and (ii) the fabrication of the gold electrodes allowing for the counting of the captured corpuscles.

The microchip consists on an arrangement of Ni concentrators with a diameter of 40 μm, height of 20 μm and spacing between them of 160 μm embedded in a silicon substrate; while 350 annular couples of gold electrodes, with thickness of 300 nm, width of 3 microns and spacing of 3 microns, are perfectly placed on top of the Ni concentrators (Figure 1f). With this configuration we maximize the probability of capturing the components in the region of the electrodes, as the maximum value of the magnetic field gradient is found at the edges of concentrators. An identical number of reference electrodes (Figure 1g) are placed nearby, in a region without magnetic concentrators underneath. The net impedance variation upon corpuscles capture on the measurement electrodes is then measured by subtracting the current flowing in the measurement and reference electrodes at fixed voltage amplitude and 1 MHz frequency.

The blood sample to be analyzed is dispensed on a glass slide where a polymeric confinement gasket, made of polydimethylsiloxane (PDMS) (Figure 1b) with thickness varying between 40 μm and 500 μm, is prefabricated. Then, a linear stepper motor lowers the microchip so that it is put in close contact with the glass slide and the confinement gasket creating the sealing that defines the fluidic cell. In order to perform the experiments at variable angle α between gravity and the chip normal, a mechanical set-up that allows to vary α between 0${}^{\circ }$ and 105${}^{\circ }$ has been realized. A motorized linear motion (Figure 1d) allows the external magnets to approach the back surface of the chip in a controlled way enabling magnetophoretic attraction. The measurement protocol is based on the magnets motion “downward” and “upward” in order to create/annihilate the field gradient which attracts/releases blood corpuscles. This allows to better disentangle the current variations proportional to the corpuscles concentration from spurious fluctuations and signal drift.

### 2.2. Experimental Configurations

Particle transport in a magnetophoretic system is influenced by many forces of different nature: (i) magnetic force due to magnetic field gradients, (ii) viscous drag, (iii) gravity, (iv) buoyancy, (v) random forces due to thermal kinetics, (vi) particle–fluid interactions, and (vii) interparticle effects such as Van der Waals force.

The gravity force ($F_{g}$), linked to the mass of the particle, and the buoyancy force ($F_{b}$), related to Archimedes’s principle, share the same direction perpendicular to the floor; however, their effects are opposite, such that the net contribution is $F_{g-b}=\frac{4}{3}\pi r_{p}^{3}\left(\rho_{p}-\rho_{fluid}\right)g$ where $\rho_{p}$ and $\rho_{fluid}$ are, respectively, the density of the particle and the fluid, $r_{p}$ the radius of the particle, and g is the gravity acceleration [26]. For a particle with radius $r_{p}$, the Drag force can be expressed as $F_{Drag}=6\pi \eta_{fluid}r_{p}\left(u-v\right)$ where $\eta_{fluid}$ is the fluid viscosity, u the velocity of the fluid, and v velocity of the one of the particles [26]. As in our device the fluid does not move, it follows that u = 0 and the classical viscous contribute of $F_{Drag}$ is opposite and linearly proportional to the velocity of the particle. The Brownian motion concerns the disorderly motion of particles arising from the collisions with fluid molecules resulting therefore in a diffusion phenomena. The average distance ($L_{diff}$) traveled by a particle in a time interval t is related to the diffusion coefficient D as $L_{diff}=∼\sqrt{Dt}$ and represents how the particle would theoretically move in that fluid without the other forces. As the diffusion coefficient D is inversely proportional to the fluid viscosity and the particle dimension, the bigger the particle and the more viscous the medium, the less will be the diffusion length. For example, the diffusion length in water after 1 s is ∼1.1 μm for hemozoin crystals (considering an average dimension of 350 nm) and ∼300 nm for red blood cells (average equivalent radius of 2.78 μm); the corresponding values in a more viscous fluid like blood are almost halved [26,27]. Therefore, considering that these values are much lower than distances covered in the same time interval due to sedimentation (typical sedimentation speeds for RBCs in blood are on the order of 2–4 μm/s), the effect of Brownian motion can be neglected. Moreover, as we are dealing with diluted particle suspensions, interparticle effects and particle/fluid interactions can also be neglected. Therefore, in the present case, as for most magnetophoretic applications involving micrometric particles, only the first four aforementioned terms are dominant and will be taken into account [26].

The test can be carried out in different configurations, exploiting the same chip: (i) a horizontal configuration with the chip surface parallel to the floor and facing downwards (Figure 2a, $\alpha =0^{\circ }$); (ii) a vertical configuration, with the chip surface perpendicular to the floor (Figure 2b, $\alpha =90^{\circ }$). It is evident that, depending on which configuration we are dealing with, forces are competing in a different way. In the horizontal case (Figure 2a), the magnetophoretic force must completely counteract the gravity force, partially compensated by the buoyancy and the drag forces, and a typical vertical capture motion is expected. In the vertical configuration, instead (Figure 2b), the magnetic force is perpendicular to gravity, thus giving rise to a more complex particle motion. Here, during sedimentation, paramagnetic particles are attracted towards the magnetic concentrators, due to the high field gradient they produce under the action of the external magnets. HC and i-RBCs are thus expected to follow curved trajectories bent towards the chip surface.

### 2.3. Estimation of the Magnetic Field Gradient Needed for Magnetophoretic Separation of Blood Corpuscles

The force exerted on a magnetic particle, assuming that the length scale over which the magnetic field varies is much bigger with respect to the size of the magnetic particle itself and that the magnetic susceptibility of the particle is constant, can be expressed as $F_{m}=\frac{1}{2}\mu_{0}V_{p}\Delta \chi \nabla H^{2}\left(X_{c}\right)$, where $V_{p}$ is the volume of the particle, Δχ = $\chi_{particle}-\chi_{fluid}$ is the difference between the magnetic susceptibility of the particle and the surrounding medium, and $H(X_{c})$ represents the field in position $X_{c}$ (i.e., center of the particle). Looking at $F_{m}$ we understand that in order to attract a particle magnetophoretically, the χ value of the particle itself is not important, but rather its difference with respect to the surrounding medium. In particular, in the experiments reported in this paper, the medium in which the blood particles are suspended is Phosphate-Buffered Saline (PBS), which has an absolute $\chi_{PBS}=-9.05\times 10^{-6}$. The relative magnetic susceptibilities Δχ of i-RBCs at the various stages (ring, thropozoite, or schizont), depending on their peculiar HC content, are listed in Table 1. In the table, Met-Hb t-RBCs represent red blood cells treated with NaNO${}_{2}$, in which hemoglobin is fully oxidized into methemoglobin, giving rise to a relative magnetic susceptibility, which is just two times that of the schizont stage. Due to the strong similarity between i-RBCs and t-RBCs, the latter have been used in this paper to mimic i-RBCs behavior in experiments carried out to optimize the diagnostic test [28].

Assuming that the volume of a red blood cells is $V_{RBC}=9.1\times 10^{-11}{\mathrm{cm}}^{-3}$, the density of an RBC is equal to $\rho_{RBC}=1.15\;g\cdot {\mathrm{cm}}^{-3}$ and the plasma density is $\rho_{p}=1.025\;g\cdot {\mathrm{cm}}^{-3}$ [29,30], the sedimentation force, calculated as the sum of gravity and buoyancy force on a treated RBC, $F_{gb}=\left(\rho_{RBC}-\rho_{P}\right)\cdot V_{RBCs}$ turns out to be $1.1\times 10^{-13}$N. According to the expression of the magnetic force on a superparamagnetic particle, $F_{m}=\frac{1}{2}\mu_{0}V\Delta \chi \nabla H^{2}$ the minimum gradient of $H^{2}$ needed to overcome the sedimentation force has been calculates and reported in Table 1. $\nabla H^{2}$ should be greater than 10${}^{15}$ A${}^{2}\cdot m^{-3}$ but lower than 10${}^{17}$ A${}^{2}\cdot m^{-3}$ in order to avoid the capture of h-RBC which would lead to false positive results. The threshold value of $\nabla H^{2}$ for HCs, instead, is much lower, on the order of $1.7\times 10^{13}A^{2}\cdot m^{-3}$, thus allowing to choose an intermediate value, slightly lower than 10${}^{15}$ A${}^{2}\cdot m^{-3}$, for a selective sorting and detection of HCs in the horizontal configuration.

This field gradient is provided macroscopically by a system of external magnets (Figure 2a,b) and microscopically by an array of nickel pillars that are thus magnetized due to the external field. The magnet assembly is made of two NdFeB (N52 grade from Supermagnete (https://www.supermagnete.it/)) permanent magnets, in the shape of parallelepipeds with 6 × 25 × 25 mm${}^{3}$ size and mechanically clamped to sandwich a μ-metal foil (0.2 mm thick) with the faces having the same polarity. With this configuration a high gradient is produced at the surface of the magnet assembly, in the order of $\nabla H^{2}$ = $7\times 10^{14}A^{2}\cdot m^{-3}$, thus fulfilling the requirement found above for the selective sorting of HCs. Note that the local $\nabla H^{2}$ at the concentrator surface in saturation is on the order of 10${}^{16}A^{2}\cdot m^{-3}$, larger than the macroscopic one produced by external magnet, thus allowing for the efficient concentration on top of measurement electrodes.

### 2.4. Red Blood Cells Treatment Protocol

As mentioned above, red blood cells can be chemically treated to change their magnetic susceptibility becoming very similar to the one measured on malaria-infected red blood cells. In particular, the treatment aims to transform the hemoglobin contained in the RBC from oxyhemoglobin (oxy-Hb), which is diamagnetic, to methemoglobin (met-Hb), which is paramagnetic. In order to cause this transformation, RBCs must be exposed to oxidizing drugs, like $N_{a}NO_{2}$ according to the following protocol. First, the whole blood sample is mixed with sodium heparin and an anticoagulant is used to avoid the blood droplets employed in the experiments to clot. Then, a centrifugation step is performed in order to separate the RBCs from both the plasma and the other blood components, such as platelets, proteins, and white blood cells. Once isolated, red blood cells are resuspended in 1 × Dulbecco’s PBS solution from Sigma-Aldrich. The resulting suspension is then oxygenated for 30 min at room temperature. The oxygenated RBCs suspension is then centrifugated and resuspended in PBS. This step is performed in order to remove RBCs that have been hemolized during the oxygenation process. After that, the sample is divided in two further samples. The former will act as a reference solution for the healthy RBCs, while the latter will undergo another step: $N_{a}NO_{2}$ solution is added to the RBC suspension in order to obtain a concentration of 840 μg/mL. The so-obtained suspension is then rocker-incubated for 30 min at room temperature and then centrifugated and resuspended in PBS at a 40% hematocrit. This will act as the starting solution for the t-RBCs from which further dilution are taken to perform the experiments reported in Section 3.

## 3. Results and Discussion

### 3.1. Test Selectivity

As anticipated above, depending on the configuration angle it is possible to tune the capture efficiency or achieve selectivity with respect to different corpuscles. Indeed, gravity force plays a fundamental role especially when α=0, in the horizontal configuration (Figure 2a). In this case, our device can perform a selective capture as the system is able to attract hemozoin crystals but not the red blood cells infected by malaria, having a volume magnetic susceptibility two orders of magnitude lower. To demonstrate this capability here we report data taken using the most recent prototype of the diagnostic apparatus, currently used also for experiments on real human blood samples. In this specific set-up, the height of the microfluidic cell is 500 microns, in order to implement an easy sample load in a cartridge.

The experiments involving hemozoin have been carried out using synthetic crystals (β-haematin) supplied by Invivogen. As a matter of fact, they have been proven to be very similar to their natural counterpart concerning the chemical structure, crystal dimension, and magnetic properties [11,32].

In Figure 2, the signals obtained from hemozoin suspension in plasma diluted 1:10 with PBS, using the chip in horizontal configuration (c) and vertical configuration, (d) are reported. The concentration of hemozoin used in this experiment is 300 ng/μL, corresponding to about $10^{7}$ HCs/μL [19]. In order to discriminate the magnetophoretic capture of HCs from a spurious drift of the signals, the magnets are disangaged in the time interval of 60–120 s in order to release captured particles and generate a detectable signal variation. Note that, both for α = 0 and 90 degrees, a sizable signal (A) is obtained superposed to a spurious drift, meaning that the system is able to capture HCs in both configurations. In particular, the amplitudes of the two signals (Figure 2c,d) are approximately equal, on the order of about 0.4 μA. This is not surprising because the $\nabla H^{2}$ experimentally produced in the set-up largely exceeds the threshold value for capture in the horizontal configuration. The capture efficiency is very similar to that of the vertical configuration, where there is no net threshold for the long-range capture by the field produced by external magnets.

For experiments on RBCs, bovine-treated red blood cells have been used to spike a suspension of untreated RBCs diluted in plasma-PBS (1:10) to obtain a total haematocrit of 4%. In case of data reported in Figure 2e,f, the t-RBC concentration was 1000 t-RBCs/μL, corresponding to a real malaria-infected blood sample with 10,000 parasites/μL diluted 1:10 in PBS, as in the actual protocol used for TMek. The microfluidic cell height for this experiment was 500 microns. In Figure 2, it is possible to notice that a sizable signal is detected only in the vertical configuration (Figure 2f), while in the horizontal configuration (Figure 2e), we just observe a small spurious signal induced by the magnet motion (at 60 s and 180 s) superposed to the signal drift, in analogy with what is found in experiments with bare PBS in the cell. The amplitude (A) of the signal in the vertical configuration (Figure 2f) is ~2 μA, while in the horizontal configuration (Figure 2e) it is approximately 200 nA.

To summarize, the experimental results reported in Figure 2 clearly show that our test is able to selectively detect only HCs in the horizontal configuration, while in the vertical one both HCs and t-RBCs are captured and detected.

### 3.2. Detection Efficiency versus the Angle between the Chip Normal and Gravity

The influence of the angle α between the chip normal and gravity (see Figure 3a) on the test performances has been studied in experiments with angles of 75 degrees, 90 degrees, and 105 degrees. In these experiments, we used a previous prototype of the diagnostic apparatus, allowing to vary both the cell height and α angle but displaying a slightly lower detection efficiency. For experiments at different angle α the cell height was fixed at 40 microns. In Figure 3b we report the measured current signal amplitude versus the angle α for: (i) a suspension of untreated RBCs in plasma-PBS (1:10 vol-vol) with haematocrit of 0.4% (red dots in Figure 3b); (ii) a synthetic model of infected blood obtained by spiking t-RBCs in the previous suspension of untreated RBCs in order to achieve a concentration of 2500 t-RBCs/μL (yellow dots in Figure 3b). The first sample has been used to evaluate the spurious signal arising from unspecific capture of untreated (healthy RBCs), that can be associated to false positive results of the diagnostic test.

As explained in Section 2.2, at 90 degrees particles tend to sediment because of the gravity, but if they are paramgnetic, as in the case of HCs or t-RBCs, the magnetic force attracts them towards the chip bending the particle motion trajectory. In this configuration, the measured current signal due to t-RBCs (and so the signal related to the i-RBCs in a real situation) is ~1.54 μA, while the non-specific signal is 0.084 μA. At 75 degrees, gravity force still has a component that opposes to the magnetic force, so that the “false positives” signal decreases, giving an amplitude A of ~0.03 μA, but also the signal related to the t-RBCs capture decreases down to about 0.037 μA. At 105 degrees the component of the gravity force contributes to bring all the blood cells, both healthy and treated, towards the silicon microchip in the region of the electrodes, so that the measured current signals related to both t-RBCs and h-RBCs increase, up to 2.91 μA and 2.55 μA, respectively.

The ratio between the specific and non-specific signal, within the error bars, is maximum for 90 degrees. Thus this is the preferred configuration in order to minimize the impact of false positive results while maximizing the test sensitivity.

### 3.3. Optimization of the Microfluidic Chamber Thickness

The height of the microfluidic chamber (δ in Figure 3a), defined by the thickness of the PDMS gasket on the glass slide, has been experimentally studied in order to maximize the measured current signal related to the capture of the t-RBCs and so the specific current signal. Figure 4 shows the amplitude of the current signal (black dots in Figure 4) measured for cell heights of 40 μm, 80 μm, and 500 μm, using the same apparatus described in last section. Surprisingly enough, the net signal decreases while increasing the thickness, despite the fact that more blood cells are present in the cell. This decreasing trend, also confirmed by multiphysics simulations (red dots in Figure 4), is related to the fact that in the considered experiments the blood sample is loaded on the PDMS cell when the cell is parallel to the ground plane ($\alpha =0^{\circ }$) so that the blood cells sediment on the glass slide within the time necessary (~90 s) to put the microchip in contact with the glass slide, make the electrical contacts, stabilize the signal, and start the measurement. Considering a typical sedimentation speed of 4 μm/s for RBCs, in the case of δ = 500 μm, a thickness of at least 360 μm from the substrate is depleted of globules. This means that when we approach the magnets near the sample, the blood cells are more distant than they are in the case of δ = 40 μm, and therefore the capture efficiency decreases.

These results are also confirmed by multiphysics simulations carried out with COMSOL Multiphysics 5.3a where each t-RBCs was modeled as a spherical particle with volume $V_{p}$ = $5.8\times 10^{-17}$$m^{3}$ and magnetic volume susceptibility with respect to PBS $\Delta \chi =3.9\times 10^{-6}$. Considering an average value of $\nabla H^{2}$ = $4\times 10^{14}A^{2}m^{-3}$ within the cell, we uniformly filled the 140 μm where RBCs amass due to sedimentation with the proper number of t-RBCs per μL corresponding to an initial concentration of 2500 t-RBCs/μL.

We can conclude that, according to the protocol adopted in this example, the best conditions for increasing the signal correspond to δ = 40 μm. Nevertheless, such a small cell height can be critical at low concentrations of target corpuscles, where the reduced volume of sample that can be analyzed can be highly detrimental to the ultimate limit of detection.

## 4. Conclusions

In this paper, we discussed some specific features of a novel system for malaria diagnosis, based on the selective magnetophoretic capture and electrical impedance detection of malaria infected red blood cells and free hemozoin crystals. We showed that the test can selectively detect free circulating hemozoin crystals or the combination of both hemozoin crystals and infected RBCs, exploiting the same chip but horizontal and vertical configurations, respectively. In the vertical configuration, we checked the influence of a fine tuning of the angle between the chip surface and gravity force around 90 degrees. We found that 90 degrees is the optimum angle allowing to achieve the largest ratio between the specific and non-specific signal, in order to minimize the impact of false positive results. The cell height also strongly influence the detection efficiency. It turns out that increasing the cell height does not allow to improve the signal at fixed concentration, mainly due to corpuscle sedimentation from the sample load to the start of the measurement, according to the measurement protocol currently used. These findings are highly relevant for future developments of the test, allowing to improve its limit of detection and reduce the impact of false positive results. Moreover, the capability of performing an automatic selective detection of hemozoin crystals and infected RBCs represent a unique feature in the arena of hemozoin based tests for malaria, which could be of great help for on-field reliable assessment of the sickness evolution.

The proposed diagnostic test, in the present form, cannot easily distinguish between real malaria-infected red blood cells and red blood cells of a patient affected by methemoglobinemia, containing a lot of methemoglobin as in treated RBCs used for the present study. Even though the incidence of methemoglobinemia with respect to malaria could be negligible in endemic zone, the interference on the test results arising from other hematic pathologies is a crucial point which will require further investigation to assess the test specificity. On the other hand, our approach could be used to diagnose also other diseases in which organisms, such as *Schistosoma mansoni* worms [33,34], feed on blood and, in turn, produce hemozoin for detoxification purposes.

## 5. Patents

“Apparatus for the quantification of biological components dispersed in a fluid” (http://hdl.handle.net/11311/1126859).

## Figures and Tables

**Figure 1 sensors-20-04972-f001:**
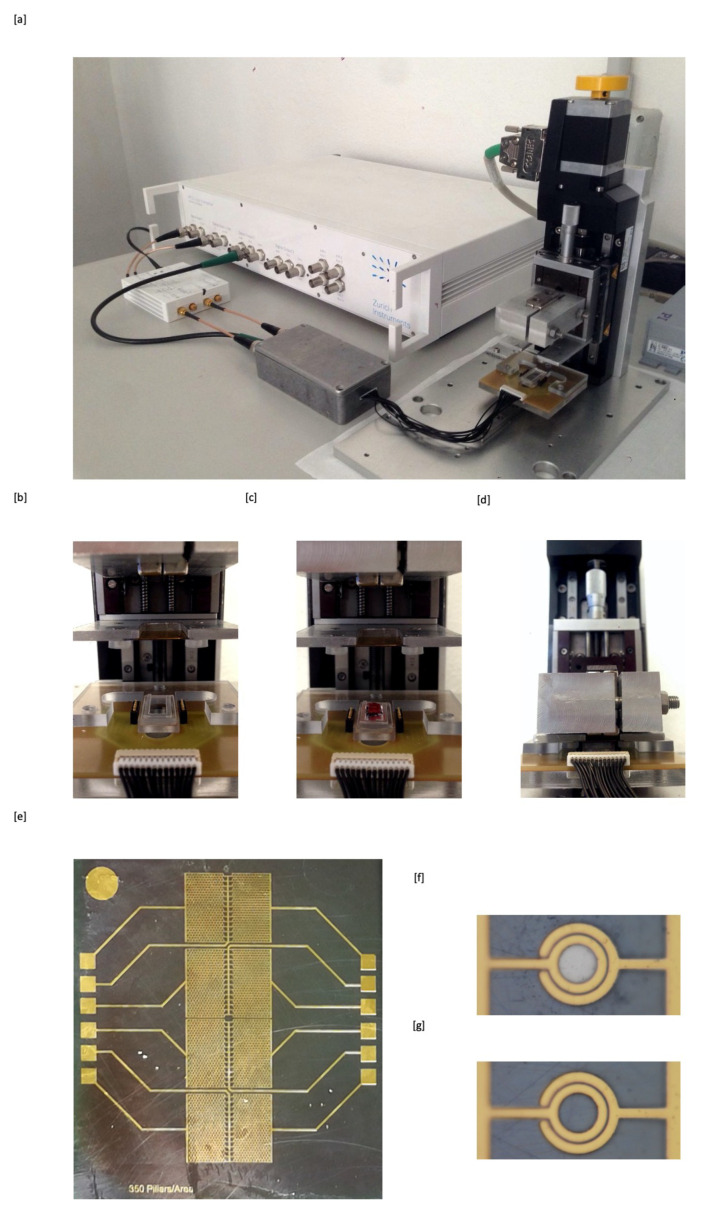
(**a**) Picture of the measurement set-up. A transimpedance amplifier (HF2TA) is used to convert the current flowing into the electrodes, upon application of a sinusoidal voltage with 100 mV amplitude at 1 MHz, in a voltage signal fed to a lock-in amplifier (HF2LI), both provided by Zurich Instrument. In this protoype, a first stepper motor (L40620SD00 from PhI) is used for the chip lowering, while a second stepper motor (M126 CG1 from PhI), mounted on the first, implements the magnet approach/disengagement. (**b**) Chip loaded on the holder mounted on the first stepper motor; (**c**) blood sample dispensed on the glass slide with a PDMS gasket; (**d**) chip pressed on the glass slide to define the fluidic chamber and realize the electric contacts on gold pads by spring contacts mounted on the same support of the glass slide; (**e**) chip layout; (**f**) zoom on a single measurement electrode on top of a Ni pillar; (**g**) zoom on a single reference electrode without Ni pillar underneath.

**Figure 2 sensors-20-04972-f002:**
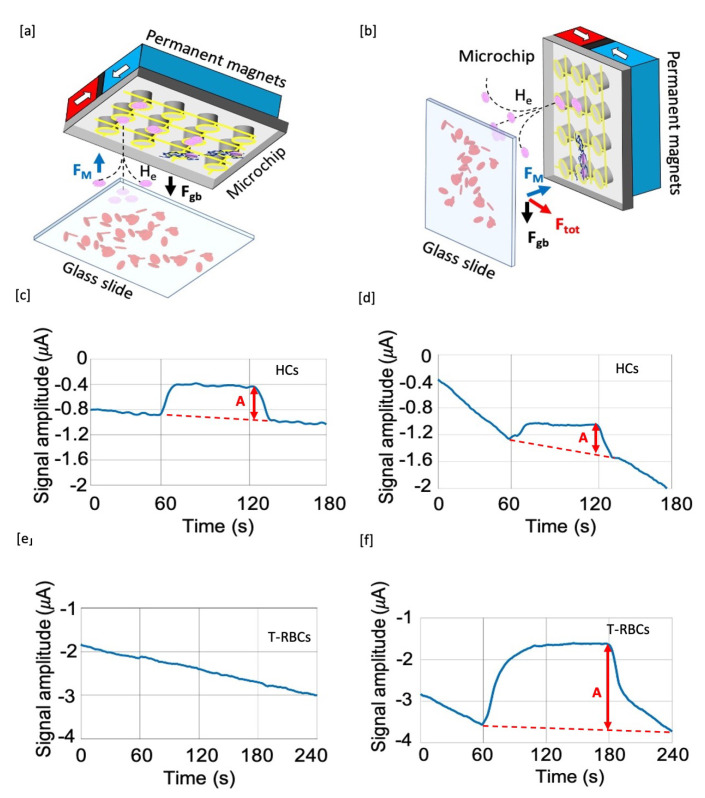
Detection system in horizontal (**a**) and vertical (**b**) configurations. In panels (**c**,**d**), the experimental measured current signal for hemozoin crystals samples is depicted, while in panels (**e**,**f**), the experimental curves are related to capture dynamics of bovine-treated red blood cells samples.

**Figure 3 sensors-20-04972-f003:**
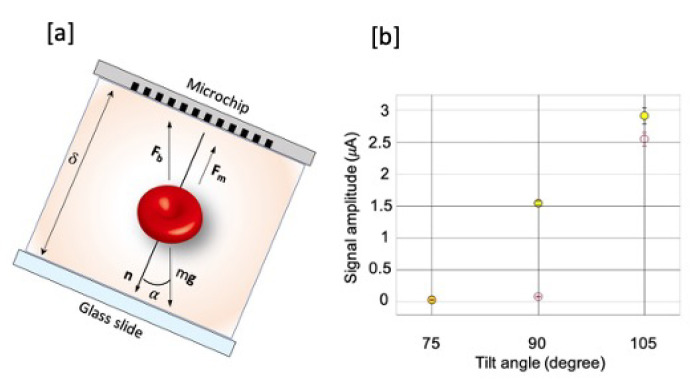
Direction of the forces in the fluidic chamber with respect to the angle configuration in (**a**) and measured current signal at different configuration angles (yellow dots) and the relative signal due to the capture of false positives in (**b**) (red empty dots).

**Figure 4 sensors-20-04972-f004:**
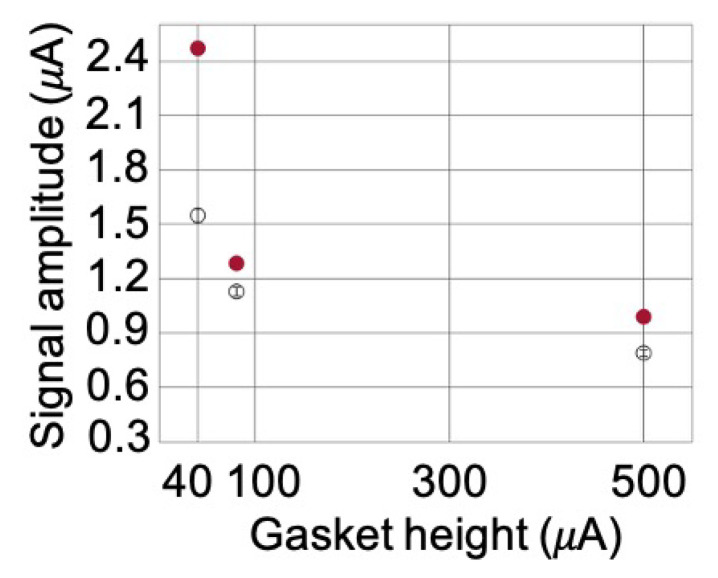
Measured current signal at different heights of the microfluidic chamber containing the blood sample (black dots) and the relative COMSOL simulation points (empty red dots). The error bars are obtained as the square root of the sum of the squares of the errors on the signal value in the 10 s before the first disengagement of the magnets and the during the 10 s before the second approach of the magnet.

**Table 1 sensors-20-04972-t001:** Net magnetic susceptibilities of hemozoin and RBCs with respect to PBS.

Corpuscle	$\Delta \chi (*10^{-6)}$	Minimum $\nabla H^{2}$ Value (A${}^{2}$/m${}^{3}$)
Healthy h-RBC [28,31]	0.01	1.56 × $10^{17}$
Ring i-RBC [31]	0.82	1.9 × $10^{15}$
Trophozoite i-RBC [31]	0.91	1.72 × $10^{15}$
Schizont i-RBC [15,17,28,31]	1.82	8.6 × $10^{14}$
met-Hb t-RBC [28]	3.9	4 × $10^{14}$
Hemozoin crystals HCs [6]	320	2.26 × $10^{13}$

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
