# Peer review of "On-Chip Selective Capture and Detection of Magnetic Fingerprints of Malaria"

_sensors, 2020, doi:10.3390/s20174972_

Round 1

Reviewer 1 Report

The manuscript entitled :''On chip selective capture and detection of magnetic
fingerprints of malaria'' seems to be a scientifically sound and throughout study of a point-of care lab on a chip portable device for malaria detection and I want to congratulate the authors.

The approach of exploiting the paramagnetic properties of the iron-containing particles (HC and infected blood cells) by varying the angle between the gravitational and the magnetic force is very ingenious. Did the authors also test the device on samples containing simultaneously both HC and t-RBC? I think the difference in mass of this twp kind of particles can also be exploited in the vertical setup keeping in mind that the heavier particles have different bending trajectories in magnetic field than the less heavy ones. 

Through the manuscript I found some small typing errors which I will point out below:

page 4 line 138  "In particular, in the experiments reported in this paper...." 

page 4 line 151 "∇H2 should be should be greater than......"

page 5 line 154 "...for a the selective..."

Author Response

Reply to referees comments on the paper On chip selective capture and detection of magnetic fingerprints of malaria” by F. Milesi et al.

Reviewer #1 (Comments and Suggestions for Authors):

The manuscript entitled :''On chip selective capture and detection of magnetic
fingerprints of malaria'' seems to be a scientifically sound and throughout study of a point-of care lab on a chip portable device for malaria detection and I want to congratulate the authors.

Reply.

We thank the Reviewer for considering TMek a scientifically sound solution and a valid lab on a chip portable device for malaria detection.

The approach of exploiting the paramagnetic properties of the iron-containing particles (HC and infected blood cells) by varying the angle between the gravitational and the magnetic force is very ingenious. Did the authors also test the device on samples containing simultaneously both HC and t-RBC? I think the difference in mass of this two kind of particles can also be exploited in the vertical setup keeping in mind that the heavier particles have different bending trajectories in magnetic field than the less heavy ones. 

Reply.

We thank the reviewer for evaluating that varying the angle is a “very ingenious” approach. The experiment proposed by the reviewer is very interesting and could open the way to a new route for disentangling signals coming from different corpuscles. This could be the subject of a dedicated publication.

Through the manuscript I found some small typing errors which I will point out below:

page 4 line 138  "In particular, in the experiments reported in this paper...." 

Reply.

We corrected the typo.

page 4 line 151 "H2 should be should be greater than......"

Reply.

We corrected the typo.

page 5 line 154 "...for a the selective..."

Reply.

We corrected the typo.

Reviewer 2 Report

A lab-on-a-chip device has been developed for the magnetic detection of malaria fingerprints in blood samples. Infected red blood cells and free hemozoin crystals are captured magnetically and detected by electrical measurements using and inter-digitated sensing electrode. Some characterizations have been performed on the device geometry and orientation. Overall, the manuscript provides novel data suitable for publication in Sensors, however, major revisions should be done in order for the manuscript to be in the standards of the journal. Bellow are my specific comments on the manuscript:

The capture and detection mechanism of the method developed in this manuscript is not well presented visually. Perhaps, an additional panel to Fig. 1 with a better zoom on the concentrator, PDMS membrane, sensing systems etc. will be more informative for the readers.

In the Experimental Configurations section further rationale and analysis should be provided why out of several forces involved, only the four of them (magnetic, viscous drag, gravity, and buoyancy) have been identified significant and others are considered negligible.

Sample preparation has not been explained. As a diagnosis/detection technique, it is important to explain the protocol by which the blood sample has been prepared. Is the intact blood sample used for the detection or the sample requires dilusion/processing steps? Further clarification should be added to the experimental section.

No standard deviation has been provided for the measurements. Does it mean there has been no replicates for the measurements? Statistical analysis should be provided to assure whether the differences are statistically significant or not.

Is the detection specific to malaria fingerprints, or other diseases might cause similar changes to RBCs or form other components with properties similar to free hemozoin crystals? Further discussions on the specificity of the developed method should be provided.

Several typos and English errors were observed in the manuscript. I recommend the authors to have the manuscript reviewed by a native English speaker.

Author Response

Reply to referees comments on the paper On chip selective capture and detection of magnetic fingerprints of malaria” by F. Milesi et al.

Reviewer #2 (Comments and Suggestions for Authors):

A lab-on-a-chip device has been developed for the magnetic detection of malaria fingerprints in blood samples. Infected red blood cells and free hemozoin crystals are captured magnetically and detected by electrical measurements using and inter-digitated sensing electrode. Some characterizations have been performed on the device geometry and orientation. Overall, the manuscript provides novel data suitable for publication in Sensors, however, major revisions should be done in order for the manuscript to be in the standards of the journal. Below are my specific comments on the manuscript:

The capture and detection mechanism of the method developed in this manuscript is not well presented visually. Perhaps, an additional panel to Fig. 1 with a better zoom on the concentrator, PDMS membrane, sensing systems etc. will be more informative for the readers.

Reply.

We thank the reviewer for the comments and revisions on the manuscript.

To clarify this point we added a new figure and related discussion in the main text. The new figure 1 shows the complete measurement setup used for the experiments presented in this paper, with a view on the electronic equipment, the stepper motors for the magnets and chip movement, the PDMS membrane where the blood sample is loaded and the chip with a zoom on the measurement (with magnetic micro-concentrator underneath) and reference electrodes (without magnetic micro-concentrators).

The modified text describing the experimental setup and chip layout, page 4, is:

“The microchip consists on an arrangement of Ni concentrators with a diameter of 40 mm height of 20 mm and spacing between them of 160 mm embedded in a silicon substrate; while 350 interdigitated gold electrodes, with thickness of 300nm, width of 3 microns and spacing of 3 microns, are perfectly placed on top of the Ni concentrators (Figure 1f). With this configuration we maximize the probability of capturing the components in the region of the electrodes, since the maximum value of the magnetic field gradient is found at the edges of the concentrators.

An identical number of reference electrodes (Figure 1g) are placed nearby, in a region without magnetic concentrators underneath. The net impedance variation upon corpuscles capture on the measurement electrodes is then measured by subtracting the current flowing in the measurement and reference electrodes at fixed voltage amplitude and 1 MHz frequency.

The blood sample to be analyzed is dispensed on a glass slide where a polymeric confinement gasket, made of Polydimethylsiloxane (PDMS) (figure 1f) with thickness varying between 40 mm and 500 mm is prefabricated.  Then a linear stepper motor lowers the microchip so that it is put in close contact with the glass slide and the confinement gasket creating the sealing that defines the fluidic cell. In order to perform the experiments in the most reproducible and automated way, a mechanical setup that allows to vary the angle alpha between 0° and 105° has been designed. A motorized linear motion (figure 1d) allows the external magnets to approach the back surface of the chip in a controlled way enabling the magnetophoretic attraction.”

In the Experimental Configurations section further rationale and analysis should be provided why out of several forces involved, only the four of them (magnetic, viscous drag, gravity, and buoyancy) have been identified significant and others are considered negligible.

Reply

To clarify this point we modified the text in the Experimental Configuration section at page 5.

The additional text is:

“The gravity force Fg linked to the mass of the particle, and the buoyancy force Fb related to Archimedes’s principle share the same direction perpendicular to the floor, however, their effects are opposite, such that the net contribute is: Fg−b=4/3πrp3p−ρfluid)g where ρp and ρfluid are, respectively, the density of the particle and the fluid, rp the radius of the particle and g is the gravity acceleration. [19] For a particle with radius rp the Drag force can be expressed as FDrag=6πhfluidrp(u−v) where hfluid is the fluid viscosity, u the velocity of the fluid while v the one of the particles. [19] Since in TMek device the fluid does not move, it follows that u = 0 and the classical viscous contribute of FDrag is opposite and linearly proportional to the velocity of the particle.

The Brownian motion concerns the disorderly motion of particles arising from the collisions with fluid molecules resulting therefore in a diffusion phenomena. The average distance (Ldiff) traveled by a particle in a time interval t is related to the diffusion coefficient D as Ldiff ~ and represents how the particle would theoretically move in that fluid without the other forces. Since the diffusion coefficient D is inversely proportional to the fluid viscosity and the particle dimension, the bigger the particle and the more viscous the medium, the less will be the diffusion length. For example, the diffusion length in water after 1 s is ~1mm for hemozoin crystals (considering an average dimension of 350nm) and ~300nm for red blood cells (average radius 2.78 mm); the corresponding values in a more viscous fluid like blood are almost halved. [19,34] Therefore, considering that these values are much lower than distances covered in the same time interval due to sedimentation (typical sedimentation speeds for RBCs in blood are on the order of 2-4μm/s) the effect of Brownian motion can be neglected.

Moreover, since we are dealing with diluted particle suspensions, interparticle effects and particle/fluid interactions can also be neglected.

So, in the present case, as for most magnetophoretic applications involving micrometric particles, only the first four aforementioned terms are dominant and will be taken into account [6].

Sample preparation has not been explained. As a diagnosis/detection technique, it is important to explain the protocol by which the blood sample has been prepared. Is the intact blood sample used for the detection or the sample requires dilusion/processing steps? Further clarification should be added to the experimental section.

Reply

For the real test application, the patient whole blood samples must be diluted 1:10 in PBS with anticoagulant to allow for efficient magnetophoretic separation. However, in the present case we use a synthetic model of malaria infected blood, made of RBCs treated with NaNO2 to obtain a magnetic behavior mimicking that of infected RBCs. To better explain the preparation of these synthetic blood samples we added a new section (“Red blood cells treatment protocol”) in the Experimental Configuration section at page 6.

The text added is:

“As mentioned above, red blood cells can be chemically treated to change their magnetic susceptibility becoming very similar to the one measured on malaria infected red blood cells. In particular, the treatment aims to transform the haemoglobin contained in the RBC from oxyhaemoglobin (oxy-Hb), which is diamagnetic, to methaemoglobin (met-Hb), which is paramagnetic. In order to cause this transformation, RBCs must be exposed to oxidising drugs, like NaNO2 according to the following protocol. Firstly, the whole blood sample is mixed with sodium heparin, an anticoagulant used to avoid the blood droplets employed in the experiments to clot.  Then a centrifugation step is performed in order to separate the RBCs from both the plasma and the other blood components, such as platelets, proteins, white blood cells. Once isolated, red blood cells are resuspended in 1x Dulbecco’s PBS (Phosphate buffer saline) solution from Sigma-Aldrich. The resulting suspension is then oxygenated for 30 min at room temperature. The oxygenated RBCs suspension is then centrifugate and resuspended in PBS. This step is performed in order to remove RBCs that have been haemolized during the oxygenation process. After that, the sample is divided in two further samples. The former will act as a reference solution for the healthy RBCs while the other one will undergo another step: NaNO2 solution is added to the RBC suspension in order to obtain a concentration of 840g/ml. The so obtained suspension is then rocker-incubated for 30 min at room temperature and then centrifugated and resuspended in PBS at a 40% haematocrit. This will act as the starting solution for the t-RBCs from which further dilution are taken to perform the experiments reported in Section 3.”

No standard deviation has been provided for the measurements. Does it mean there has been no replicates for the measurements? Statistical analysis should be provided to assure whether the differences are statistically significant or not.

Reply

Error bars for each experimental point are now shown in Figures 3 and Figure 4. Experiments for each configuration have been repeated at least twice, by repeated cycles of magnets approach/removal in which the amplitude is measured as indicated in Figure 2. As a matter of fact, we obtain a consistent trend by looking at signals measured in correspondence of the same cycle, so that in Figure 3,4 we plot the amplitude of the signal variation corresponding to the first release. The overall error for this amplitude estimate is evaluated as follows: the standard deviation of the raw signal is calculated in the 10 seconds before the first disengagement of the magnets, and the same is done when the signal variation due to release is fully accomplished, within the 10 seconds before the second approach of the magnet. The error Aerr associated to the estimation of the amplitude of the signal is obtained by taking the square root of the sum of the squares of the two standard deviations.

Is the detection specific to malaria fingerprints, or other diseases might cause similar changes to RBCs or form other components with properties similar to free hemozoin crystals? Further discussions on the specificity of the developed method should be provided.

To answer the reviewer’s question, we added a paragraph in the conclusion section at page 9 as follow:

“The proposed diagnostic test, in the present form, cannot easily distinguish between real malaria infected red blood cells and red blood cells of a patient affected by methemoglobinemia, containing a lot of methaemoglobin as in treated RBCs used for the present study. Even though the incidence of methaemoglobinemia with respect to malaria could be negligible in endemic zone, the interference on the test results arising from other hematic pathologies is a crucial point which will require further investigation to assess the test specificity. On the other hand, TMek approach could be used to diagnose also other diseases in which organisms, such as Schistosoma Mansoni worms [38,39], feed on blood and, in turn, produce hemozoin for detoxification purposes.”

Several typos and English errors were observed in the manuscript. I recommend the authors to have the manuscript reviewed by a native English speaker.

Thanks for the suggestion

Reviewer 3 Report

The manuscript by Milesi et al. described a quantitative and rapid detection method for malaria-infected red blood cells (i-RBCs) and free hemozoin crystals depending on their paramagnetic properties. The authors previously reported the electrical and magnetic properties of hemozoin nanocrystals (Appl. Phys. Lett., 2018, 113, 203703).

Overall, the manuscript contains valuable information, but the experimental details are missing in the manuscript, so that the method described does not allow the readers to repeat the experiments.

In particular, the detailed structure of Ni microconcentrators and 350 interdigitated gold electrodes are not shown in Figure 1A and B. The impedance variation on the electrodes is measured by subtracting the current flowing in the measurement and reference electrode at fixed voltage amplitude and 1 MHz frequency. Unfortunately, Figure 1 does not provide any information on the reference electrode. Many experimental setups, including instruments and motorized linear motion setup, are not clear. In this sense, the detailed device format of the TMek test is required in the manuscript.   

The main results focused only on the performance comparison depending on the configuration of the chip, which can affect the selective magnetophoretic separation in the developed “TMek” test system.

Why does the chip configuration affect the TMek selectivity between hemozoin crystal and t-RBCs? There is no discussion about the results. In addition, the detection efficiency is not compared to other measurement systems and real infected RBCs samples with false-positive results.

In addition, there are many typesetting errors in the manuscript.

In page 2, line 37: “on the on-field compatible version” is not “LAMP”. LAMP should be spelled out in the manuscript. In the same way, a novel rapid diagnostic test is not an abbreviation of TMek.

In Figure 1, it should be read as the bovine-treated red blood cell (t-RBC). All abbreviations and acronyms in the text must be defined the first time used.

Note that there is a space between the number and the unit.

Author Response

Reply to referees comments on the paper On chip selective capture and detection of magnetic fingerprints of malaria” by F. Milesi et al.

Reviewer #3 (Comments and Suggestions for Authors):

The manuscript by Milesi et al. described a quantitative and rapid detection method for malaria-infected red blood cells (i-RBCs) and free hemozoin crystals depending on their paramagnetic properties. The authors previously reported the electrical and magnetic properties of hemozoin nanocrystals (Appl. Phys. Lett., 2018, 113, 203703).

Overall, the manuscript contains valuable information, but the experimental details are missing in the manuscript, so that the method described does not allow the readers to repeat the experiments.

In particular, the detailed structure of Ni micro-concentrators and 350 interdigitated gold electrodes are not shown in Figure 1A and B. The impedance variation on the electrodes is measured by subtracting the current flowing in the measurement and reference electrode at fixed voltage amplitude and 1 MHz frequency. Unfortunately, Figure 1 does not provide any information on the reference electrode. Many experimental setups, including instruments and motorized linear motion setup, are not clear. In this sense, the detailed device format of the TMek test is required in the manuscript.   

 Reply

We thank the reviewer for the comments and revisions on the manuscript.

To clarify this point we added a new figure and related discussion in the main text. The new figure 1 shows the complete measurement setup used for the experiments presented in this paper, with a view on the electronic equipment, the stepper motors for the magnets and chip movement, the PDMS membrane where the blood sample is loaded and the chip with a zoom on the measurement (with magnetic micro-concentrator underneath) and reference electrodes (without magnetic micro-concentrators).

The microchip consists on an arrangement of Ni concentrators with a diameter of 40 mm height of 20 mm and spacing between them of 160 mm embedded in a silicon substrate; while 350 interdigitated gold electrodes, with thickness of 300nm, width of 3 microns and spacing of 3 microns, are perfectly placed on top of the Ni concentrators (Figure 1f). With this configuration we maximize the probability of capturing the components in the region of the electrodes, since the maximum value of the magnetic field gradient is found at the edges of the concentrators.

An identical number of reference electrodes (Figure 1g) are placed nearby, in a region without magnetic concentrators underneath. The net impedance variation upon corpuscles capture on the measurement electrodes is then measured by subtracting the current flowing in the measurement and reference electrodes at fixed voltage amplitude and 1 MHz frequency.

The blood sample to be analyzed is dispensed on a glass slide where a polymeric confinement gasket, made of Polydimethylsiloxane (PDMS) (figure 1f) with thickness varying between 40 mm and 500 mm is prefabricated.  Then a linear stepper motor lowers the microchip so that it is put in close contact with the glass slide and the confinement gasket creating the sealing that defines the fluidic cell. In order to perform the experiments in the most reproducible and automated way, a mechanical setup that allows to vary the angle alpha between 0° and 105° has been designed. A motorized linear motion (figure 1d) allows the external magnets to approach the back surface of the chip in a controlled way enabling the magnetophoretic attraction.”

The main results focused only on the performance comparison depending on the configuration of the chip, which can affect the selective magnetophoretic separation in the developed “TMek” test system.

Why does the chip configuration affect the TMek selectivity between hemozoin crystal and t-RBCs?

 Reply

As explained in section 3.1, TMek setup can work in two configurations: a first horizontal configuration, in which the surface of the chip is placed parallel to the tabletop, and a vertical configuration, where the surface of the chip is placed perpendicular to the table top. It is evident that, depending on which configuration we are dealing with, forces are competing in a different way, and this is displayed in Figure 2. Gravity force plays a fundamental role especially when α = 0, in the horizontal configuration (Figure 2 a). In this case, TMek can perform a selective capture since the system is able to attract hemozoin crystals but not the red blood cells infected by malaria, having a volume magnetic susceptibility two orders of magnitude lower. Indeed, the gradient that is produced at the surface of the magnet assembly is in the order of  ∇H2=7·1014 A2 · m−3  and it largely exceeds the threshold value of ∇H2 for HCs that is on the order of 1.7 · 1013 A2 · m−3 but not the threshold value of ∇H2 for i-RBCs  that is on the order of 1015 A2 · m−3.

To better clarify this point we report the lines in the text:

“∇H2 should be greater than 1015A2·m−3 but lower than 1017A2·m−3 in order to avoid the capture of h-RBC which would lead to false positive results. The threshold value of ∇H2 for HCs, instead, is much lower, on the order of 1.7·1013A2·m−3, thus allowing to choose an intermediate value, slightly lower than 1015A2·m−3, for a selective sorting and detection of HCs in the horizontal configuration.”

And also

“With this configuration a high gradient is produced at the surface of the magnet assembly, in the order of ∇H2=7·1014A2·m−3, thus fulfilling the requirement found above for the selective sorting of HCs.”

There is no discussion about the results. In addition, the detection efficiency is not compared to other measurement systems and real infected RBCs samples with false-positive results.

 Reply

We discuss our experimental results by comparing them with Multiphysics simulations in Figure 4 and providing an explanation of the detection efficiency variation as a function of the cell height at page 9, where we say:

“This decreasing trend, also confirmed by Multiphysics simulations (red dots in Figure 4) is related to the fact that in the considered experiments the blood sample is loaded on the PDMS cell when the cell is parallel to the ground plane (alpha=0°) so that the blood cells sediment on the glass slide within the time necessary (about 90 s) to put the microchip in contact with the glass slide, make the electrical contacts, stabilize the signal and start the measure. Considering a typical sedimentation speed of 4 mm for RBCs, in the case of d= 500 mm, a thickness of at least 360 mm from the substrate is depleted of globules. This means that when we approach the magnets near the sample, the blood cells are more distant than they are in the case of d= 40 mm, and therefore the capture efficiency decreases.”

Concerning the absolute comparison of the detection efficiency with other diagnostic tests, we fully understand the referee request, but this is definitely beyond the scope of the present paper, dealing with the investigation of some specific parameters on the test response, using synthetic models of malaria infected red blood cells. A full comparison with other diagnostic tests would require a statistically powered analysis in endemic zone, on hundreds of patients, which is a major step forward with respect to the present laboratory investigation.

In addition, there are many typesetting errors in the manuscript.

In page 2, line 37: “on the on-field compatible version” is not “LAMP”. LAMP should be spelled out in the manuscript. In the same way, a novel rapid diagnostic test is not an abbreviation of TMek.

Reply.

We corrected the text according to referee’s suggestions: the new text is: “known as Loop-Mediated Isothermal Amplification or LAMP” and “In this paper we discuss the capability of a novel rapid diagnostic test called TMek we have recently introduced”.

In Figure 1, it should be read as the bovine-treated red blood cell (t-RBC). All abbreviations and acronyms in the text must be defined the first time used.

Note that there is a space between the number and the unit.

 Reply.

We corrected the typos.

Round 2

Reviewer 2 Report

All of my comments have been addressed and I have no further comments.

Reviewer 3 Report

Compared to the previous version, the revised manuscript improved the quality and presentation of the work, including experimental details.
Thus, I would recommend for publication in Sensors.